# Explaining Black-box Model Predictions via Two-level Nested Feature Attributions with Consistency Property

## Abstract

Techniques that explain the predictions of black-box machine learning models are crucial to make the models transparent, thereby increasing trust in AI systems. The input features to the models often have a nested structure that consists of high- and low-level features, and each high-level feature is decomposed into multiple low-level features. For such inputs, both high-level feature attributions (HiFAs) and low-level feature attributions (LoFAs) are important for better understanding the model's decision. In this paper, we propose a model-agnostic local explanation method that effectively exploits the nested structure of the input to estimate the two-level feature attributions simultaneously. A key idea of the proposed method is to introduce the consistency property that should exist between the HiFAs and LoFAs, thereby bridging the separate optimization problems for estimating them. Thanks to this consistency property, the proposed method can produce HiFAs and LoFAs that are both faithful to the black-box models and consistent with each other, using a smaller number of queries to the models. In experiments on image classification in multiple instance learning and text classification using language models, we demonstrate that the HiFAs and LoFAs estimated by the proposed method are accurate, faithful to the behaviors of the black-box models, and provide consistent explanations.

## 1 Introduction

The rapid increase in size and complexity of machine learning (ML) models has led to a growing concern about their *black-box* nature. Models provided as cloud services are literal black boxes, as users have no access to the models themselves and the training data used. This opacity raises numerous concerns, including issues of trust, accountability, and transparency. Consequently, techniques to explain the predictions made by those black-box models have been attracting significant attention (Danilevsky et al., 2020; Došilović et al., 2018; Saeed & Omlin, 2023).

Various *model-agnostic* local explanation methods have been proposed to explain the predictions of black-box models. The representative methods are, for example, local interpretable model-agnostic explanation (LIME) (Ribeiro et al., 2016) and kernel Shapley additive explanations (Kernel SHAP) (Lundberg & Lee, 2017), which estimate the feature attributions of the individual prediction by approximating the model's behavior with local linear surrogate models around the input.

In LIME and Kernel SHAP, the input to the model is generally assumed to be a flat structure, where the input features are treated as independent variables. In many realistic tasks for various domains, such as image, text, geographic, e-commerce, and social network data, however, the input features have a nested structure that consists of high- and low-level features, and each high-level feature is decomposed into multiple low-level features. A typical task with such nested features is multiple instance learning (MIL) (Ilse et al., 2018) where the model is formulated as set functions (Kimura et al., 2024). In MIL, the input is a set of instances, the high-level feature is an instance in the set, and the low-level features represent the features of the instance. In addition, even if the input is not represented with a nested structure when it is fed into the model, it may be more natural to interpret it with the nested structure. For example, although a text input is usually represented as a sequence

Figure 1: Example of the black-box model prediction for a nested structured input and its corresponding high- and low-level feature attributions estimated by the proposed method with consistency constraints. Objects in each high-level feature represent the low-level features.

of words, it is natural to interpret it as having high-level features such as phrases, sentences, and paragraphs.

The two-level features enable us to understand the model predictions with two types of feature attributions that have different levels of granularity in explanation, which we name *high-level feature attributions (HiFAs)* and *low-level feature attributions (LoFAs)*, respectively. Figure 1 shows an example of the prediction for a nested structured input and its corresponding HiFAs and LoFAs. The HiFAs represent how much each of the high-level features contributes to the prediction. These are also referred to as *instance attributions* in the MIL literature (Early et al., 2022; Javed et al., 2022), which are used to reveal which instances strongly affected the model's decision. On the other hand, the LoFAs represent how much each of the low-level features contributes to the prediction, providing a more fine-grained explanation of how the components of the instances affected the prediction. Both the HiFAs and LoFAs are important for understanding the model's decision. *However, existing studies have focused on estimating either-level attributions, and no study has addressed estimating the HiFAs and LoFAs simultaneously.*

For the estimation of the HiFAs and LoFAs, two naive approaches can be applied. One is to estimate the HiFAs and LoFAs separately by applying existing model-agnostic local explanation methods to the high- and low-level features, respectively. The other is to estimate the LoFAs first, as with the former approach, and then estimate the HiFAs by aggregating the LoFAs. However, these approaches have two rooms for improvement in terms of using the nested structure of the input. First, even though the queries to the black-box model are often limited in practice due to the computational time and request costs, the input structure is not utilized to reduce the number of queries in the estimation. Second, the former approach can produce inconsistent explanations between the HiFAs and LoFAs, for example, the most influential high-level feature and the high-level feature to which the most influential low-level feature belongs may not match.

To address these issues, we propose a model-agnostic local explanation method that effectively exploits the nested structure of the input to estimate the HiFAs and LoFAs simultaneously. A key idea of the proposed method is to introduce the consistency property that should exist between the HiFAs and LoFAs, thereby bridging the separate optimization problems for them. We solve a joint optimization problem to estimate the HiFAs and LoFAs simultaneously with the consistency constraints depicted in Figure 1 based on the alternating direction method of multipliers (ADMM) (Boyd et al., 2011). The algorithm is a general framework that can also introduce various types of regularizations and constraints for the HiFAs and LoFAs, such as the $\ell_1$ and $\ell_2$ regularizations and non-negative constraints, which lead to the ease of interpretability for humans.

In experiments, we quantitatively and qualitatively assess the HiFAs and LoFAs estimated by the proposed method on image classification in the MIL setting and text classification using language models, compared with estimating them separately and using a recent attribution method for MIL (Early et al., 2022). The experimental results show that the HiFAs and LoFAs estimated by the proposed method 1) satisfy the consistency property, 2) are faithful explanations to the black-box models even when the number of queries to the model is small, 3) can accurately guess the ground-truth positive instances and their features in the MIL task, and 4) are reasonable explanations visually.

The contributions of this work are summarized as follows:

- This study is the first to propose a model-agnostic local explanation method to estimate the two-level nested feature attributions simultaneously, which satisfies the consistency property between them.

- In the experiments on practical tasks, we demonstrated that the proposed method could produce accurate, faithful, and consistent two-level feature attributions with a smaller number of queries to the black-box models.

## 2 RELATED WORK

Numerous methods for explaining the individual predictions of black-box models have been proposed in the literature (Ribeiro et al., 2016; Lundberg & Lee, 2017; Ribeiro et al., 2018; Petsiuk et al., 2018; Plumb et al., 2018). A versatile approach is to explain feature attributions estimated by approximating the model predictions with surrogate models around the input, such as LIME (Ribeiro et al., 2016) and Kernel SHAP (Lundberg & Lee, 2017). The proposed method is in line with this type of approach.

Set data is one of the nested input features, which treats a set of multiple instances as a single input. Set data appears in various ML applications, such as point cloud classification (Guo et al., 2021), medical image analysis (Cheplygina et al., 2019), and group recommendation (Dara et al., 2020), and the explainability on those applications has also been studied in the literature (Tan & Kotthaus, 2022; van der Velden et al., 2022). Unlike our work, most such studies focus only on estimating instance attributions corresponding to those of high-level features. For example, Early et al. proposed to estimate instance attributions by learning surrogate models with MIL-suitable kernel functions (Early et al., 2022).

Several studies have addressed estimating feature attributions effectively by leveraging group information of input features. In the natural language processing literature, some studies estimated sentence- and phrase-level feature attributions by grouping words in the same sentence and phrase together and regarding them as a single feature (Zafar et al., 2021; Mosca et al., 2022). In addition, Rychener et al. showed that word-level feature attributions can be improved by generating perturbations at a sentence level, mitigating the issues of out-of-distribution for the model and high-dimensional search space (Rychener et al., 2023). In the official SHAP library (shap (Github), 2024), by grouping input features by hierarchical clustering in advance and generating perturbations at the group level, one can reduce the number of queries to the model.

If we consider high-level features as nodes and low-level features as the features of the nodes and then somehow put edges between the nodes, we can think of an input as a graph. By doing so, model-agnostic explanation methods for graphs, such as GNNExplainer (Ying et al., 2019) and GraphLIME (Huang et al., 2023), can be applied to our task. However, since this approach highly relies on the graph structure, additional information is required to create appropriate edges.

## 3 PROPOSED METHOD

### 3.1 TWO-LEVEL NESTED FEATURE ATTRIBUTIONS WITH SURROGATE MODELS

The model $f$ to be explained is a trained black-box model that takes an arbitrary input, such as tabular, image and text, $\boldsymbol{x} \in \mathcal{X}$, and outputs a prediction $\boldsymbol{y} = f(\boldsymbol{x}) \in [0,1]^C$ where $\mathcal{X}$ is the input space and $C$ is the number of classes. The input $\boldsymbol{x}$ is made of two-level nested features, referred to as high-level and low-level features, and the high-level feature is decomposed into multiple low-level features. In particular, the input $\boldsymbol{x}$ is represented as a set or sequence of $J$ high-level features, i.e., $\boldsymbol{x} = \{\boldsymbol{x}_j\}_{j=1}^J$ where $\boldsymbol{x}_j \in \mathbb{R}^{D_j}$ is the $D_j$-dimensional low-level feature vector representing the $j$-th high-level feature. One example of such input appears in image classification under the MIL setting. In this setting, the input is a bag of images, the high-level feature is an image in the bag, and the low-level features correspond to super-pixels in the image. Another example appears in a document classification where the input is a sequence of sentences, the high-level feature is a sentence in the sequence, and the low-level features correspond to the words in the sentence.

We consider estimating the high-level feature attributions (HiFAs) and low-level feature attributions (LoFAs) that explain the prediction of the black-box model $f$ for the input $\boldsymbol{x}$ using surrogate models as with LIME and Kernel SHAP. The HiFAs and LoFAs represent how much high- and low-level features in the input contribute to the prediction, respectively. In the aforementioned MIL setting, the HiFAs represent how much images in the input bag contribute to the prediction, which is also referred to as *instance attributions* in the literature, and the LoFAs represent how much super-pixels in the images contribute to the prediction. To estimate the HiFAs and LoFAs, we introduce two-level local linear surrogate models for high-level and low-level features, $e^{\mathrm{H}}$ and $e^{\mathrm{L}}$, that mimic the behaviors of the black-box model $f$ around the input $\boldsymbol{x}$, as follows:

$$e^{\mathrm{H}}(\boldsymbol{z}^{\mathrm{H}};\boldsymbol{\alpha}) = \sum_{j=1}^{J} \alpha_j z_j^{\mathrm{H}}, \qquad e^{\mathrm{L}}(\boldsymbol{z}^{\mathrm{L}};\boldsymbol{\beta}) = \sum_{j=1}^{J} \sum_{d=1}^{D_j} \beta_{jd} z_{jd}^{\mathrm{L}}, \tag{1}$$

where $\boldsymbol{z}^{\mathrm{H}} \in \{0,1\}^J$ and $\boldsymbol{z}^{\mathrm{L}} = \{\boldsymbol{z}_j^{\mathrm{L}}\}_{j=1}^{J}$ with $\boldsymbol{z}_j^{\mathrm{L}} \in \{0,1\}^{D_j}$ are simplified inputs associated with the input $\boldsymbol{x}$, which are used to indicate the presence or absence of the high- and low-level features in $\boldsymbol{x}$, respectively; $\boldsymbol{\alpha} \in \mathbb{R}^J$ and $\boldsymbol{\beta} = \{\boldsymbol{\beta}_j\}_{j=1}^{J}$ with $\boldsymbol{\beta}_j \in \mathbb{R}^{D_j}$ are the learnable coefficients of these surrogate models, and after learning, they will be the HiFAs and LoFAs themselves, respectively. For ease of computation below, we define the concatenation of $\boldsymbol{\beta}$ and $\boldsymbol{z}^{\mathrm{L}}$ over the high-level features as $\boldsymbol{\beta}^{\dagger} = \mathtt{concat}(\boldsymbol{\beta}_1, \boldsymbol{\beta}_2, \cdots, \boldsymbol{\beta}_J) \in \mathbb{R}^{D^{\dagger}}$ and $\boldsymbol{z}^{\mathrm{L}\dagger} = \mathtt{concat}(\boldsymbol{z}_1^{\mathrm{L}}, \boldsymbol{z}_2^{\mathrm{L}}, \cdots, \boldsymbol{z}_J^{\mathrm{L}}) \in \{0,1\}^{D^{\dagger}}$, where $D^{\dagger} = \sum_{j=1}^{J} D_j$.

The surrogate models are learned with the predictions of the black-box model $f$ for perturbations around the input $\boldsymbol{x}$. The perturbations are generated by sampling the simplified inputs $\boldsymbol{z}^{\mathrm{H}}$ and $\boldsymbol{z}^{\mathrm{L}\dagger}$ from binary uniform distributions and then constructing masked inputs $\phi_{\boldsymbol{x}}^{\mathrm{H}}(\boldsymbol{z}^{\mathrm{H}}), \phi_{\boldsymbol{x}}^{\mathrm{L}}(\boldsymbol{z}^{\mathrm{L}\dagger}) \in \mathcal{X}$ depending on the simplified inputs, respectively. Here, $\phi_{\boldsymbol{x}}^{\mathrm{H}}$ and $\phi_{\boldsymbol{x}}^{\mathrm{L}}$ are mask functions that replace the input $\boldsymbol{x}$'s dimensions associated with the dimensions being zero in the simplified inputs $\boldsymbol{z}^{\mathrm{H}}$ and $\boldsymbol{z}^{\mathrm{L}\dagger}$ with uninformative values, such as zero, respectively. Let $\boldsymbol{Z}^{\mathrm{H}} \in \{0,1\}^{N_{\mathrm{H}} \times J}$ and $\boldsymbol{Z}^{\mathrm{L}} \in \{0,1\}^{N_{\mathrm{L}} \times D^{\dagger}}$ be the matrices whose rows are the generated simplified inputs for the high- and low-level features, respectively, where $N_{\mathrm{H}}$ and $N_{\mathrm{L}}$ are the numbers of perturbations used to estimate the HiFAs and LoFAs, respectively. Also, let $\tilde{\boldsymbol{y}}^{\mathrm{H}} = [\tilde{y}_1^{\mathrm{H}}, \tilde{y}_2^{\mathrm{H}}, \cdots, \tilde{y}_{N_{\mathrm{H}}}^{\mathrm{H}}]^{\top} \in \mathbb{R}^{N_{\mathrm{H}} \times C}$ and $\tilde{\boldsymbol{y}}^{\mathrm{L}} = [\tilde{y}_1^{\mathrm{L}}, \tilde{y}_2^{\mathrm{L}}, \cdots, \tilde{y}_{N_{\mathrm{L}}}^{\mathrm{L}}]^{\top} \in \mathbb{R}^{N_{\mathrm{L}} \times C}$ be the predictions of the black-box model for the perturbations where $\tilde{y}_n^{\mathrm{H}} = f(\phi_{\boldsymbol{x}}^{\mathrm{H}}(\boldsymbol{Z}_n^{\mathrm{H}}))$ and $\tilde{y}_n^{\mathrm{L}} = f(\phi_{\boldsymbol{x}}^{\mathrm{L}}(\boldsymbol{Z}_n^{\mathrm{L}}))$.

Simply, the parameters of the surrogate models, i.e., the HiFAs $\hat{\boldsymbol{\alpha}}$ and LoFAs $\hat{\boldsymbol{\beta}}^{\dagger}$, can be estimated by solving the following weighted least squares separately:

$$\hat{\boldsymbol{\alpha}} = \underset{\boldsymbol{\alpha}}{\arg\min}\, \mathcal{L}_{\mathrm{H}}(\boldsymbol{\alpha}) + \lambda_{\mathrm{H}}\Omega_{\mathrm{H}}(\boldsymbol{\alpha}) \quad \text{where} \quad \mathcal{L}_{\mathrm{H}}(\boldsymbol{\alpha}) = \frac{1}{2}(\tilde{\boldsymbol{y}}^{\mathrm{H}} - \boldsymbol{Z}^{\mathrm{H}}\boldsymbol{\alpha})^{\top}\boldsymbol{W}^{\mathrm{H}}(\tilde{\boldsymbol{y}}^{\mathrm{H}} - \boldsymbol{Z}^{\mathrm{H}}\boldsymbol{\alpha}), \tag{2}$$

$$\hat{\boldsymbol{\beta}}^{\dagger} = \underset{\boldsymbol{\beta}^{\dagger}}{\arg\min}\, \mathcal{L}_{\mathrm{L}}(\boldsymbol{\beta}^{\dagger}) + \lambda_{\mathrm{L}}\Omega_{\mathrm{L}}(\boldsymbol{\beta}^{\dagger}) \quad \text{where} \quad \mathcal{L}_{\mathrm{L}}(\boldsymbol{\beta}^{\dagger}) = \frac{1}{2}(\tilde{\boldsymbol{y}}^{\mathrm{L}} - \boldsymbol{Z}^{\mathrm{L}}\boldsymbol{\beta}^{\dagger})^{\top}\boldsymbol{W}^{\mathrm{L}}(\tilde{\boldsymbol{y}}^{\mathrm{L}} - \boldsymbol{Z}^{\mathrm{L}}\boldsymbol{\beta}^{\dagger}),$$

$$\tag{3}$$

where $\boldsymbol{W}^{\mathrm{H}} \in \mathbb{R}^{N_{\mathrm{H}} \times N_{\mathrm{H}}}$ and $\boldsymbol{W}^{\mathrm{L}} \in \mathbb{R}^{N_{\mathrm{L}} \times N_{\mathrm{L}}}$ are the diagonal matrices whose $n$th diagonal elements represent the sample weights for the $n$th perturbation; $\Omega_{\mathrm{H}}$ and $\Omega_{\mathrm{L}}$ are the regularizers for the HiFAs and LoFAs, respectively; $\lambda_{\mathrm{H}} \geq 0$ and $\lambda_{\mathrm{L}} \geq 0$ are the regularization strengths.

### 3.2 Joint Optimization with Consistency Constraints

Although the HiFAs and LoFAs provide different levels of explanations, these explanations for the same black-box model should be consistent between them. From the linearity of the surrogate models and the fact that each high-level feature can be decomposed into low-level features, the following property is expected to be satisfied:

**Property 1** (Consistency between two-level feature attributions)**.**

$$\alpha_j = \sum_{d=1}^{D_j} \beta_{jd} \quad (\forall j \in [J]). \tag{4}$$

The two surrogate models (1) satisfying the consistency property behave equivalently for the simplified inputs $z^H$ and $z^L$ such that if $z_j^H = 0$ then $z_j^L = \mathbf{0}_{D_j}$, and if $z_j^H = 1$ then $z_j^L = \mathbf{1}_{D_j}$ where $\mathbf{0}_{D_j}$ and $\mathbf{1}_{D_j}$ are the $D_j$-dimensional zero and one vectors, respectively.

The consistency property is essential to provide consistent and convincing explanations to humans. However, it is often not satisfied for two reasons in practice. First, the number of perturbations is insufficient to accurately estimate the feature attributions because the number of queries to the model $f$ is often limited due to the computational time and request costs. Second, in the predictions for the perturbations, the behaviors of the model $f$ can differ between when the high-level features are masked out and when the low-level ones are masked out due to *missingness bias* (Jain et al., 2022). To overcome these problems, the proposed method estimates the HiFAs and LoFAs simultaneously by solving the following optimization with consistency constraints:

$$\hat{\boldsymbol{\alpha}}, \hat{\boldsymbol{\beta}}^{\dagger} = \underset{\boldsymbol{\alpha}, \boldsymbol{\beta}^{\dagger}}{\operatorname{argmin}} \, \mathcal{L}_H(\boldsymbol{\alpha}) + \mathcal{L}_L(\boldsymbol{\beta}^{\dagger}) + \lambda_H \Omega_H(\boldsymbol{\alpha}) + \lambda_L \Omega_L(\boldsymbol{\beta}^{\dagger}) \quad \text{s.t.} \quad \alpha_j = \sum_{d=1}^{D_j} \beta_{jd} \quad (\forall j \in [J]).$$

(5)

The consistency constraints bridge the two surrogate models, forcing them to behave equivalently. This helps complement the insufficiency of the queries to the model and mitigate the negative effects of the missingness bias on the estimation of the HiFAs and LoFAs.

We solve the optimization based on the alternating direction method of multipliers (ADMM) (Boyd et al., 2011). The detailed derivation of the optimization algorithm is provided in Appendix A. An advantage of employing the ADMM is that despite the interdependence of $\boldsymbol{\alpha}$ and $\boldsymbol{\beta}^{\dagger}$ caused by the consistency constraints, they can be estimated independently as in (2) and (3). In addition, the solution has another merit in that we can implement various types of regularizations and constraints for $\boldsymbol{\alpha}$ and $\boldsymbol{\beta}$, such as sparse regularization and non-negative constraints in $\Omega_H$ and $\Omega_L$. In this paper, we instantiate the proposed method with the LIME-like formulation, that is, we use the cosine kernel for calculating the sample weights $W^H$ and $W^L$ and the $\ell_2$ regularization for $\Omega_H$ and $\Omega_L$. The optimization algorithm for this instantiation is provided in Algorithm 1 in Appendix A.

**Computational Complexity.** In the proposed method, the dominant computation cost is brought by the predictions of the black-box model $f$ for the perturbations, whose computational time complexity is $O((N_H + N_L)Q)$ where $N_H$ and $N_L$ are the numbers of perturbations for the HiFAs and LoFAs, respectively, and $Q$ is the computational time complexity of $f$ in prediction once. $Q$ is often large when executing large models and models provided as cloud services. Therefore, estimating the HiFAs and LoFAs accurately with small $N_H$ and $N_L$ is crucial. In the experiments in Section 4, we demonstrate that the proposed method can estimate high-quality HiFAs and LoFAs even when $N_H$ and $N_A$ are small. A detailed discussion on the computational time complexity is provided in Appendix B.

## 4 EXPERIMENTS

We conducted experiments on two tasks, image classification in an MIL setting and text classification using language models, to evaluate the effectiveness of the proposed method, referred to as *Consistent Two-level Feature Attribution (C2FA)*. In the experiments, we implemented the proposed method in Algorithm 1 in Appendix A. Its hyperparameters, $\lambda_H$, $\lambda_L$, and $\mu_1$, were tuned using the validation subset of each dataset within the following ranges: $\lambda_H, \lambda_L \in \{0.1, 1\}$, and $\mu_2 \in \{0.001, 0.01, 0.1\}$. The remaining hyperparameters were set to $\mu_1 = 0.1$, $\epsilon_1 = \epsilon_2 = 10^{-4}$, respectively. All the experiments were conducted on a server with an Intel Xeon Gold 6148 CPU and an NVIDIA Tesla V100 GPU.

**Comparing Methods.** As comparing methods, we used the following five methods, named LIME (Ribeiro et al., 2016), MILLI (Early et al., 2022), Bottom-Up LIME (BU-LIME), Top-Down LIME (TD-LIME), and Top-Down MILLI (TD-MILLI). With LIME, we estimated the HiFAs and LoFAs separately by solving (2) and (3), respectively, where we used the cosine kernel for the sample weights and $\ell_2$ regularization for $\Omega_H$ and $\Omega_L$. Hence, LIME can be regarded as the proposed method without the consistency constraints. MILLI is the state-of-the-art instance attribution method in the MIL setting, which was proposed for estimating the HiFAs only. Therefore, we estimated the LoFAs

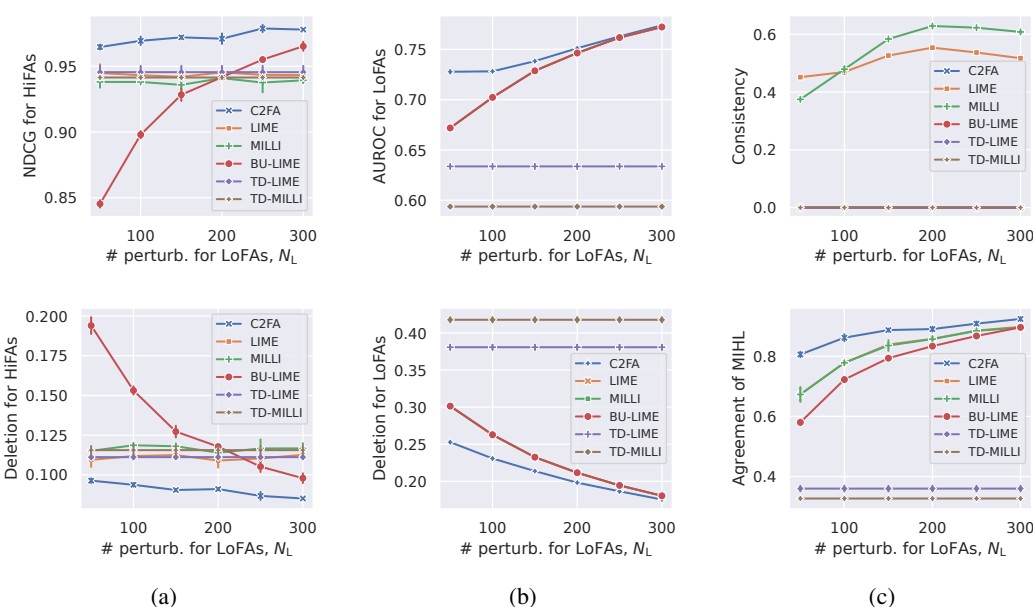

(a)          (b)          (c)

Figure 2: Quantitative evaluation on the image classification task. (a) NDCG (higher is better) and deletion scores (lower is better) of the estimated HiFAs. (b) AUROC (higher is better) and deletion scores (lower is better) of the estimated LoFAs. (c) Consistency scores (lower is better) and the agreement scores of MIHL (higher is better). The error bars represent the standard deviations of the scores over three runs with different random seeds.

in MILLI as with LIME. With BU-LIME, we first estimated the LoFAs using LIME and then calculated the HiFAs of each high-level feature by summing the LoFAs associated with the high-level feature. This method always satisfies the consistency property because the HiFAs are calculated from the LoFAs. With TD-LIME and TD-MILLI, we first estimated the HiFAs using LIME and MILLI, respectively. Then, for the $j$th high-level feature, we determined the FAs associated with it, $\boldsymbol{\beta}_j$, with the samples from the normal distribution with the mean of the $j$th HiFA $\alpha_j$ and the standard deviation of $1/D_j$. Finally, by selecting the $d$th low-level feature at random and replacing it with $\beta_{jd} = \alpha_j - \sum_{d' \in [D_j] \setminus \{d\}} \beta_{jd'}$, we obtained the LoFAs associated with the $j$th high-level feature such that they satisfy the consistency property.

## 4.1 IMAGE CLASSIFICATION IN MULTIPLE INSTANCE LEARNING

**Dataset.** We constructed an MIL dataset from the Pascal VOC semantic segmentation dataset (Everingham et al., 2015) that allows us to evaluate the estimated HiFAs and LoFAs with the ground-truth instance- and pixel-level labels. With the training subset of the dataset, each sample (bag) has from three to five images (high-level features) drawn at random from the training subset of the Pascal VOC. Here, low-level features correspond to regions (super-pixels) of each image, which are obtained by the quick shift algorithm (Vedaldi & Soatto, 2008). Each bag is labeled positive if at least an image in the bag is associated with "cat" label and negative otherwise. Also, each image pixel is labeled positive if the pixel is associated with "cat" label and negative otherwise. We used the instance- and pixel-level supervision only for evaluation. Similarly, we constructed validation and test subsets whose samples contain images from the training and test subsets of the Pascal VOC, respectively. The number of samples in training, validation, and test subsets is 5,000, 1,000, and 2,000, respectively, and the positive and negative samples ratio is equal.

**Black-box Model.** We used DeepSets permutation-invariant model (Zaheer et al., 2017) with ResNet-50 (He et al., 2016) as black-box model $f$ to be explained. We describe the implementation details of the model in Appendix C.1. Here, the test accuracy of the model was 0.945.

**Quantitative Evaluation.** We assessed the estimated HiFAs and LoFAs in terms of correctness, faithfulness, and consistency. The correctness is evaluated using the ground-truth instance- and pixel-level labels. Following the evaluation in the MIL study (Early et al., 2022), we evaluated the estimated HiFAs with normalized discounted cumulative gain (NDCG). For the estimated LoFAs, as with the evaluation of the LoFAs for single image classification (Sampaio & Cordeiro, 2023), we evaluated them as the predictions of the pixel-level labels by the area under ROC curve (AUROC) in the binary semantic segmentation manner. In the faithfulness evaluation, we assessed whether the estimated HiFAs and LoFAs are faithful to the behaviors of the model $f$ based on insertion and deletion metrics. The insertion and deletion metrics evaluate the change in the predictions of the model $f$ when features deemed important in the LoFAs are gradually added and removed from the sample, respectively (Petsiuk et al., 2018). In our experiments, we gradually add and remove the low-level features across all the high-level features in descending order of their LoFAs. Also, for the HiFAs, we add and remove the high-level features instead of the low-level ones, respectively. In terms of the consistency evaluation, we used the following two metrics. The first one is the consistency between the estimated HiFAs and LoFAs, which is calculated with $\|\boldsymbol{\alpha} - \boldsymbol{M}\boldsymbol{\beta}^\dagger\|^2$ used to calculate the penalty for the consistency constraints in (7). The second one is the agreement of the most important high- and low-level feature (MIHL), which is calculated by the ratio that the high-level feature of the highest HiFA is identical to the one associated with the low-level feature of the highest LoFA.

We evaluated the above metrics using only the samples with the positive bag label because we could not evaluate the correctness of those with the negative bag label. We ran the evaluations three times with different random seeds and reported the average scores and their standard deviation.

### 4.1.1 Results

Figure 2a shows the NDCG and deletion scores of the estimated HiFAs over various numbers of perturbations for the LoFAs, $N_{\mathrm{L}}$, where we fixed the number of perturbations for the HiFAs, $N_{\mathrm{H}} = 5$. We found that the proposed method (C2FA) consistently achieved the best NDCG and deletion scores, and the superiority of the proposed method is especially noticeable when $N_{\mathrm{L}}$ is small. Although BU-LIME improved the scores as $N_{\mathrm{L}}$ increased, the scores were still lower than those of the proposed method. Since the other comparing methods estimate the HiFAs without the effects of the LoFAs, their scores were constant regardless of the value of $N_{\mathrm{L}}$. In Appendix C.2, we show that similar results were obtained in terms of the insertion metric. In addition, when we fixed $N_{\mathrm{H}} = 20$, the methods other than BU-LIME equally achieved the highest NDCG and insertion scores regardless of $N_{\mathrm{L}}$ because $N_{\mathrm{H}}$ was sufficiently large to estimate the HiFAs accurately.

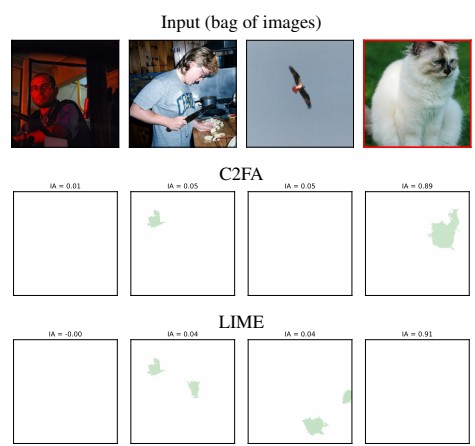

Figure 3: Example of the estimated HiFAs and LoFAs on the image classification task when $N_{\mathrm{H}} = 20$ and $N_{\mathrm{L}} = 50$. The input is shown on the first row, where the image with the red border is the positive instance. The LoFAs of super-pixels estimated by the proposed method and LIME are shown on the second and third rows, respectively, where the green color's intensity indicates the magnitude of the LoFA.

Figure 2b shows the AUROC and deletion scores of the estimated LoFAs over various values of $N_{\mathrm{L}}$ where we fixed $N_{\mathrm{H}} = 20$. When $N_{\mathrm{L}}$ is small, we found that the proposed method significantly achieved the highest AUROC and deletion scores. In particular, the AUROC score of the proposed method at $N_{\mathrm{L}} = 50$ was much the same as those of the second-best methods, LIME, MILLI, and BU-LIME, at $N_{\mathrm{L}} = 150$, and the deletion score of the proposed method at $N_{\mathrm{L}} = 50$ was much the same as that of the second-best methods at $N_{\mathrm{L}} = 100$. These results show that the proposed method is very efficient for the number of queries to the model $f$ owing to the simultaneous estimation of the HiFAs and LoFAs.

Figure 2c shows the consistency scores and the agreement scores of MIHL over various values of $N_{\mathrm{L}}$ where we fixed $N_{\mathrm{H}} = 20$. Here, the consistency scores of BU-LIME, TD-LIME, and TD-MILLI are always zero by definition. We found that the consistency scores of LIME and MILLI were worse because they estimated the HiFAs and LoFAs separately. On the other hand, those of the proposed method were nearly zero, which means that the estimated HiFAs and LoFAs satisfied the consistency property. With the agreement scores of MIHL, we found that the proposed method outperformed the other methods regardless of the values of $N_{\mathrm{L}}$, and the differences in the scores were especially noticeable at the small $N_{\mathrm{L}}$ values, i.e., $N_{\mathrm{L}} \leq 150$.

We visualize an example of the estimated HiFAs and LoFAs by the proposed method and the best-comparing method, LIME, in Figure 3. Here, we only display the LoFAs larger than 0.1 for ease of understanding. The figure shows that the proposed method assigned a high LoFA to the super-pixel in the high-level feature with the positive and highest HiFA (HiFA = 0.89), although LIME assigned high LoFAs to the super-pixels in the negative instances. The critical difference between the two methods is whether the HiFAs and LoFAs are estimated simultaneously or separately. Since both the proposed method and LIME assigned the highest HiFA to the positive instance correctly, the result indicates that estimating the HiFAs and LoFAs simultaneously is effective. Similar results were obtained in other examples shown in Appendix C.3.

## 4.2 Text Classification Using Language Models

Another practical application of the proposed method is to explain the attributions of sentences and the words they contain in text classification with language models.

**Dataset.** For evaluation, we constructed a dataset whose validation and test subsets are made of 500 and 1,000 product review texts extracted randomly from the training and test subsets of the Amazon reviews dataset (Zhang et al., 2015), respectively. Each sample in the dataset is made of multiple sentences regarded as high-level features, where each sentence is represented as a sequence of words regarded as low-level features, and the sample label represents the review's polarity, positive or negative.

**Black-box Model.** To simulate access to black-box language models provided as cloud services, we experimented using BERT (Devlin et al., 2018) with the weights fine-tuned on the original Amazon reviews dataset, which is provided on Hugging Face (fabriceyhc , Hugging Face). The test accuracy of the model is 0.947. When masking a word in the input to generate perturbed inputs, we replaced the word with the predefined mask token `[MASK]`. Similarly, when masking a sentence, we replaced all the words in the sentence with the mask token.

**Quantitative Evaluation.** Because no ground-truth labels for HiFAs and LoFAs are available in the dataset, we evaluated the estimated HiFAs and LoFAs only in terms of faithfulness and consistency, as with Section 4.1.

### 4.2.1 Results

Figure 4a shows the deletion scores of the estimated HiFAs and LoFAs over various values of $N_{\mathrm{L}}$ where we fixed $N_{\mathrm{H}} = 5$ and 50, respectively. With the deletion scores of the HiFAs, although the scores of the proposed method were equal to or worse than those of MILLI and TD-MILLI at $N_{\mathrm{L}} \leq 150$, the proposed method achieved the best at $N_{\mathrm{L}} \geq 200$. We found that in this task, the LIME-based methods, including the proposed method, were worse than the MILLI-based methods at the small $N_{\mathrm{L}}$ values. As $N_{\mathrm{L}}$ increased, the proposed method benefited from the consistency constraints and became the only LIME-based method that outperformed the MILLI-based methods. In Appendix D.1, we show that similar results were obtained in terms of the insertion metric, and when we fixed $N_{\mathrm{H}} = 50$, the scores did not change regardless of the values of $N_{\mathrm{L}}$ because $N_{\mathrm{H}}$ was sufficiently large to estimate the HiFAs accurately. With the deletion scores of the LoFAs, the proposed method outperformed the other methods regardless of the values of $N_{\mathrm{L}}$.

Figure 4b shows the consistency scores and the agreement scores of MIHL over various values of $N_{\mathrm{L}}$ where we fixed $N_{\mathrm{H}} = 50$. Again, in this task, the consistency scores of the proposed method were nearly zero regardless of the values of $N_{\mathrm{L}}$. With the agreement scores of MIHL, the proposed method kept high scores regardless of the values of $N_{\mathrm{L}}$, although the scores of BU-LIME were slightly better than the proposed method at $N_{\mathrm{L}} \leq 100$.

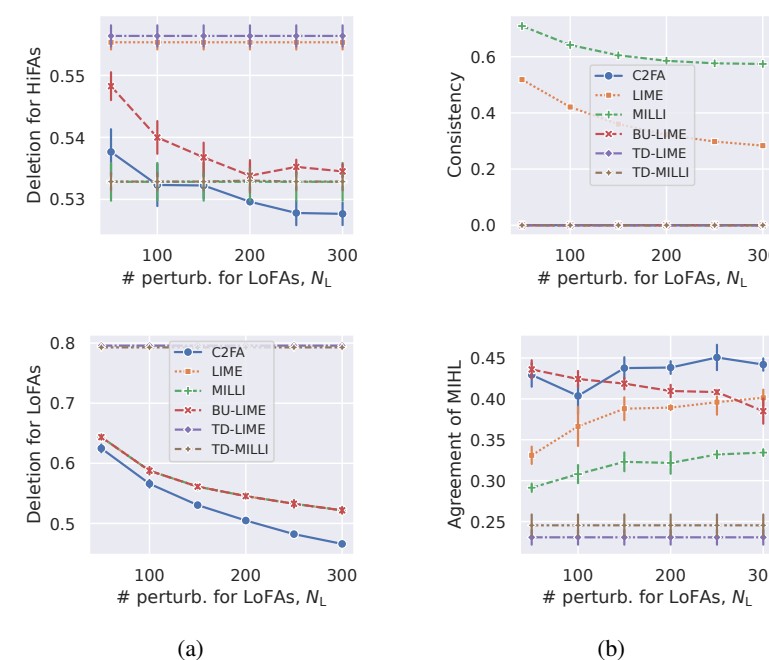

(a)  (b)

Figure 4: Quantitative evaluation on the text classification task. (a) Deletion scores of the estimated HiFAs and the estimated LoFAs (lower is better). (b) Consistency scores (lower is better) and the agreement scores of MIHL (higher is better).

Figure 5 shows an example of the HiFAs and LoFAs estimated by the proposed method and the second-best method, BU-LIME. In the example, we fixed at $N_{\mathrm{H}} = 50$ and $N_{\mathrm{L}} = 50$; that is, $N_{\mathrm{L}}$ is insufficient to estimate the LoFAs accurately. We found that although the comparing method assigned higher LoFAs to the words in the second sentence (S2), the proposed method assigned higher LoFAs to the words in the first sentence (S1). This result is because the proposed method can regularize the LoFAs by exploiting the fact that S1 has a high HiFA via the consistency constraints. Other examples are shown in Appendix D.3.

## 5 LIMITATIONS AND BROADER IMPACTS

A possible limitation of the proposed method is that the quality of the HiFAs and LoFAs may be worse in cases where the consistency property is inherently not satisfied. For example, they may happen when the HiFAs and LoFAs are estimated with the combination of different approaches, such as MILLI and LIME, and when the behaviors of the black-box model vary significantly between perturbed inputs that high- and low-level features are partially masked. To detect such an undesirable situation early, monitoring the losses of the surrogate models, $\mathcal{L}_{\mathrm{H}}$ in (2) and $\mathcal{L}_{\mathrm{L}}$ in (3), is effective because they are likely to be worse even if the objective (5) is minimized.

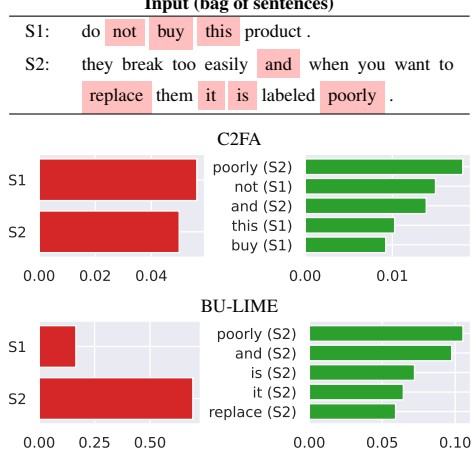

Figure 5: Example of the estimated HiFAs and LoFAs for a negative review text when $N_{\mathrm{H}} = 50$ and $N_{\mathrm{L}} = 50$. The review text is shown at the top, and the HiFAs (left) and the top-5 highest LoFAs (right) estimated by each method are shown at the bottom. Here, the words on the pink background in the review text are those appearing in the chart of the LoFAs.

Our work contributes to improving the transparency of black-box models. However, it should be noted that high-quality feature attributions may give hints about stealing the information that the model's providers want to hide, such as the training data and the model's decision-making process. To prevent such risks, it is essential to establish guidelines that ensure that the feature attributions are not used for malicious purposes.

## 6 CONCLUSION

We proposed a model-agnostic local explanation method for nested structured inputs, which explains two-level feature attributions, referred to as HiFAs and LoFAs, simultaneously. We hypothesized that the consistency property naturally derived from the characteristics of the surrogate models is essential to produce explanations that are accurate, faithful and consistent between HiFAs and LoFAs with a smaller number of queries to the model. Then, we presented an optimization algorithm that estimates the HiFAs and LoFAs while forcing them to ensure the consistency property. We demonstrated that the proposed method can produce high-quality explanations query-efficiently in the experiments on image classification in multiple instance learning and text classification using large language models. In future work, we will expand the applicability of the proposed method by extending it to tasks with three or more levels of nested features, such as multi-multi instance learning (Tibo et al., 2020).

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
