^{\mathrm{H}}$ and $z^{\mathrm{L}}$ such that if $z_j^{\mathrm{H}} = 0$ then $z_j^{\mathrm{L}} = \mathbf{0}_{D_j}$, and if $z_j^{\mathrm{H}} = 1$ then $z_j^{\mathrm{L}} = \mathbf{1}_{D_j}$ where $\mathbf{0}_{D_j}$ and $\mathbf{1}_{D_j}$ are the $D_j$-dimensional zero and one vectors, respectively.

The consistency property is essential to provide consistent and convincing explanations to humans. However, it is often not satisfied for two reasons in practice. First, the number of perturbations is insufficient to accurately estimate the feature attributions because the number of queries to the model $f$ is often limited due to the computational time and request costs. Second, in the predictions for the perturbations, the behaviors of the model $f$ can differ between when the high-level features are masked out and when the low-level ones are masked out due to *missingness bias* (Jain et al., 2022). To overcome these problems, the proposed method estimates the HiFAs and LoFAs simultaneously by solving the following optimization with consistency constraints:

$$\hat{\boldsymbol{\alpha}}, \hat{\boldsymbol{\beta}}^\dagger = \underset{\boldsymbol{\alpha}, \boldsymbol{\beta}^\dagger}{\operatorname{argmin}} \, \mathcal{L}_{\mathrm{H}}(\boldsymbol{\alpha}) + \mathcal{L}_{\mathrm{L}}(\boldsymbol{\beta}^\dagger) + \lambda_{\mathrm{H}} \Omega_{\mathrm{H}}(\boldsymbol{\alpha}) + \lambda_{\mathrm{L}} \Omega_{\mathrm{L}}(\boldsymbol{\beta}^\dagger) \quad \text{s.t.} \quad \alpha_j = \sum_{d=1}^{D_j} \beta_{jd} \quad (\forall j \in [J]). \tag{5}$$

The consistency constraints bridge the two surrogate models, forcing them to behave equivalently. This helps complement the insufficiency of the queries to the model and mitigate the negative effects of the missingness bias on the estimation of the HiFAs and LoFAs.

We solve the optimization based on the alternating direction method of multipliers (ADMM) (Boyd et al., 2011). The detailed derivation of the optimization algorithm is provided in Appendix A. An advantage of employing the ADMM is that despite the interdependence of $\boldsymbol{\alpha}$ and $\boldsymbol{\beta}^\dagger$ caused by the consistency constraints, they can be estimated independently as in (2) and (3). In addition, the solution has another merit in that we can implement various types of regularizations and constraints for $\boldsymbol{\alpha}$ and $\boldsymbol{\beta}$, such as sparse regularization and non-negative constraints in $\Omega_{\mathrm{H}}$ and $\Omega_{\mathrm{L}}$. In this paper, we instantiate the proposed method with the LIME-like formulation, that is, we use the cosine kernel for calculating the sample weights $\boldsymbol{W}^{\mathrm{H}}$ and $\boldsymbol{W}^{\mathrm{L}}$ and the $\ell_2$ regularization for $\Omega_{\mathrm{H}}$ and $\Omega_{\mathrm{

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

## A  OPTIMIZATION ALGORITHM

Our objective to estimate the HiFAs and LoFAs simultaneously is as follows:

$$
\hat{\boldsymbol{\alpha}}, \hat{\boldsymbol{\beta}}^{\dagger} = \underset{\boldsymbol{\alpha}, \boldsymbol{\beta}^{\dagger}}{\operatorname{argmin}} \, \mathcal{L}_{\mathrm{H}}(\boldsymbol{\alpha}) + \mathcal{L}_{\mathrm{L}}(\boldsymbol{\beta}^{\dagger}) + \lambda_{\mathrm{H}} \Omega_{\mathrm{H}}(\boldsymbol{\alpha}) + \lambda_{\mathrm{L}} \Omega_{\mathrm{L}}(\boldsymbol{\beta}^{\dagger}) \quad \text{s.t.} \quad \alpha_j = \sum_{d=1}^{D_j} \beta_{jd} \quad (\forall j \in [J]).
\tag{6}
$$

We solve the optimization based on the alternating direction method of multipliers (ADMM) (Boyd et al., 2011). By introducing auxiliary variables $\bar{\boldsymbol{\alpha}} \in \mathbb{R}^J$ and $\bar{\boldsymbol{\beta}}^{\dagger} \in \mathbb{R}^{D^{\dagger}}$ and Lagrange multipliers $\boldsymbol{v}_1 \in \mathbb{R}^J$, $\boldsymbol{v}_2 \in \mathbb{R}^{D^{\dagger}}$, and $\boldsymbol{v}_3 \in \mathbb{R}^J$ based on the ADMM manner, our objective is rewritten as follows:

$$
\begin{aligned}
\hat{\boldsymbol{\alpha}}, \hat{\boldsymbol{\beta}}^{\dagger} = \underset{\boldsymbol{\alpha}, \boldsymbol{\beta}^{\dagger}}{\operatorname{argmin}} \, & \mathcal{L}_{\mathrm{H}}(\boldsymbol{\alpha}) + \mathcal{L}_{\mathrm{L}}(\boldsymbol{\beta}^{\dagger}) + \lambda_{\mathrm{H}} \Omega_{\mathrm{H}}(\bar{\boldsymbol{\alpha}}) + \lambda_{\mathrm{L}} \Omega_{\mathrm{L}}(\bar{\boldsymbol{\beta}}^{\dagger}) \\
& + \boldsymbol{v}_1^{\top} \boldsymbol{h}_1(\boldsymbol{\alpha}, \bar{\boldsymbol{\alpha}}) + \boldsymbol{v}_2^{\top} \boldsymbol{h}_2(\boldsymbol{\beta}^{\dagger}, \bar{\boldsymbol{\beta}}^{\dagger}) + \boldsymbol{v}_3^{\top} \boldsymbol{h}_3(\boldsymbol{\alpha}, \boldsymbol{\beta}^{\dagger}) \\
& + \frac{\mu_1}{2} \left\{ \|\boldsymbol{h}_1(\boldsymbol{\alpha}, \bar{\boldsymbol{\alpha}})\|^2 + \|\boldsymbol{h}_2(\boldsymbol{\beta}^{\dagger}, \bar{\boldsymbol{\beta}}^{\dagger})\|^2 \right\} \\
& + \frac{\mu_2}{2} \|\boldsymbol{h}_3(\boldsymbol{\alpha}, \boldsymbol{\beta}^{\dagger})\|^2,
\end{aligned}
\tag{7}
$$

where $\boldsymbol{h}_1(\boldsymbol{\alpha}, \bar{\boldsymbol{\alpha}}) = \boldsymbol{\alpha} - \bar{\boldsymbol{\alpha}}$, $\boldsymbol{h}_2(\boldsymbol{\beta}^{\dagger}, \bar{\boldsymbol{\beta}}^{\dagger}) = \boldsymbol{\beta}^{\dagger} - \bar{\boldsymbol{\beta}}^{\dagger}$, $\boldsymbol{h}_3(\boldsymbol{\alpha}, \boldsymbol{\beta}^{\dagger}) = \boldsymbol{\alpha} - \boldsymbol{M} \boldsymbol{\beta}^{\dagger}$. Here, $\boldsymbol{M} \in \{0, 1\}^{J \times D^{\dagger}}$ is a binary matrix to add up the LoFAs associated with the same high-level feature where we set to $M_{jd} = 1$ if the $d$th feature of the concatenated input $\boldsymbol{x}^{\dagger}$ belongs to the $j$th high-level feature $\boldsymbol{x}_j$, and $M_{jd} = 0$ otherwise. The hyperparameters $\mu_1 \geq 0$ and $\mu_2 \geq 0$ are the penalty parameters for the regularization and the consistency constraint, respectively.

The optimization of (7) is performed by alternating the updates of the variables iteratively. We summarize the optimization algorithm using $\ell_2$ regularization for $\Omega_{\mathrm{H}}$ and $\Omega_{\mathrm{L}}$ in Algorithm 1. Here, $\boldsymbol{I}_J$ and $\boldsymbol{I}_{D^{\dagger}}$ are identity matrices of size $J$ and $D^{\dagger}$, respectively, and in Line 9, the variables at the zeroth step are initialized with zero. The algorithm is terminated when $\|\bar{\boldsymbol{\alpha}}^{t-1} - \bar{\boldsymbol{\alpha}}^t\|^2 + \|\bar{\boldsymbol{\beta}}^{\dagger t-1} - \bar{\boldsymbol{\beta}}^{\dagger t}\|^2 < \epsilon_1$ and $\|\boldsymbol{h}_1(\boldsymbol{\alpha}^t, \bar{\boldsymbol{\alpha}}^t)\|^2 + \|\boldsymbol{h}_2(\boldsymbol{\beta}^{\dagger t}, \bar{\boldsymbol{\beta}}^{\dagger t})\|^2 + \|\boldsymbol{h}_3(\boldsymbol{\alpha}^t, \boldsymbol{\beta}^{\dagger t})\|^2 < \epsilon_2$ where $\epsilon_1, \epsilon_2 \geq 0$ are hyperparameters. The other hyperparameters of the algorithm are $\lambda_{\mathrm{H}}, \lambda_{\mathrm{L}}, \mu_1$, and $\mu_2$.

## B  COMPUTATIONAL TIME COMPLEXITY

The computational time complexity of Algorithm 1 is split into three parts. The first part is the predictions for the perturbed inputs (Line 3), which is $O((N_{\mathrm{H}} + N_{\mathrm{L}})Q)$ where $Q$ is the computational cost of the model $f$ in prediction. The second part is the pre-computation before the iterations (Lines 4–8), which is $O(J^3 + D^{\dagger 3} + J^2 N_{\mathrm{H}} + J N_{\mathrm{H}}^2 + D^{\dagger 2} N_{\mathrm{L}} + D^{\dagger} N_{\mathrm{L}}^2)$. The third part is the iterations (Lines 11–20), which is $O(TJ^2 D^{\dagger} + TJD^{\dagger 2})$ where $T$ is the number of iterations. Compared with estimating the HiFAs and LoFAs by solving (2) and (3) separately, the third part is an additional computational cost in the proposed method. However, because one wants to execute the model on low-resource devices and cloud services, $Q$ is often large; consequently, the first part could be dominant. Therefore, estimating the HiFAs and LoFAs accurately with small $N_{\mathrm{H}}$ and $N_{\mathrm{L}}$, i.e., small amounts of perturbed inputs, is crucial in practical situations. In the experiments in Section 4, we demonstrate that the proposed method can estimate high-quality HiFAs and LoFAs even when $N_{\mathrm{H}}$ and $N_{\mathrm{L}}$ are small. In Appendix D.2, we show that the actual computational time scales linearly with $N_{\mathrm{H}}$ and $N_{\mathrm{L}}$ as with estimating the HiFAs and LoFAs separately.

---

**Algorithm 1** Estimating consistent two-level feature attributions (C2FA) with $\ell_2$ regularization

---

1: Generate binary random matrices $\boldsymbol{Z}^{\mathrm{H}}$ and $\boldsymbol{Z}^{\mathrm{L}}$

2: Obtain perturbed inputs $\{\tilde{\boldsymbol{X}}_n^{\mathrm{H}}\}_{n=1}^{N_{\mathrm{H}}}$ and $\{\tilde{\boldsymbol{X}}_n^{\mathrm{L}}\}_{n=1}^{N_{\mathrm{L}}}$ using $\phi_{\mathrm{H}}$ and $\phi_{\mathrm{L}}$

3: Obtain predictions $\tilde{\boldsymbol{y}}^{\mathrm{H}}$ and $\tilde{\boldsymbol{y}}^{\mathrm{L}}$ from the perturbed inputs

4: Obtain weight matrices $\boldsymbol{W}^{\mathrm{H}}$ and $\boldsymbol{W}^{\mathrm{L}}$

5: $\boldsymbol{A} = (\boldsymbol{Z}^{\mathrm{H}^\top} \boldsymbol{W}^{\mathrm{H}} \boldsymbol{Z}^{\mathrm{H}} + (\mu_1 + \mu_2)\boldsymbol{I}_J)^{-1}$

6: $\boldsymbol{B} = \boldsymbol{A}\boldsymbol{Z}^{\mathrm{H}^\top} \boldsymbol{W}^{\mathrm{H}} \tilde{\boldsymbol{y}}^{\mathrm{H}}$

7: $\boldsymbol{C} = (\boldsymbol{Z}^{\mathrm{L}^\top} \boldsymbol{W}^{\mathrm{L}} \boldsymbol{Z}^{\mathrm{L}} + \mu_1 \boldsymbol{I}_{D^\dagger} + \mu_2 \boldsymbol{M}^\top \boldsymbol{M})^{-1}$

8: $\boldsymbol{D} = \boldsymbol{C}\boldsymbol{Z}^{\mathrm{L}^\top} \boldsymbol{W}^{\mathrm{L}} \tilde{\boldsymbol{y}}^{\mathrm{L}}$

9: Initialize $\boldsymbol{\alpha}^0, \bar{\boldsymbol{\alpha}}^0, \boldsymbol{\beta}^{\dagger 0}, \bar{\boldsymbol{\beta}}^{\dagger 0}, \boldsymbol{v}_1^0, \boldsymbol{v}_2^0, \boldsymbol{v}_3^0$ with zero

10: $t = 0$

11: **repeat**

12:     $\boldsymbol{\alpha}^{t+1} = \boldsymbol{B} + \boldsymbol{A}(\mu_2 \boldsymbol{M} \boldsymbol{\beta}^{\dagger t} - \mu_1 \bar{\boldsymbol{\alpha}}^t - \boldsymbol{v}_1^t - \boldsymbol{v}_3^t)$

13:     $\bar{\boldsymbol{\alpha}}^{t+1} = (\mu_1 + 2\lambda_{\mathrm{H}})^{-1}(\boldsymbol{v}_1^t + \mu_1 \boldsymbol{\alpha}^{t+1})$

14:     $\boldsymbol{\beta}^{\dagger t+1} = \boldsymbol{D} + \boldsymbol{C}(\boldsymbol{M}^\top \boldsymbol{v}_3^t + \mu_1 \bar{\boldsymbol{\beta}}^{\dagger t} + \mu_2 \boldsymbol{M}^\top \boldsymbol{\alpha}^{t+1} - \boldsymbol{v}_2^t)$

15:     $\bar{\boldsymbol{\beta}}^{\dagger t+1} = (\mu_1 + 2\lambda_{\mathrm{L}})^{-1}(\boldsymbol{v}_2^t + \mu_1 \boldsymbol{\beta}^{\dagger t+1})$

16:     $\boldsymbol{v}_1^{t+1} = \boldsymbol{v}_1^t + \mu_1(\boldsymbol{\alpha}^{t+1} - \bar{\boldsymbol{\alpha}}^{t+1})$

17:     $\boldsymbol{v}_2^{t+1} = \boldsymbol{v}_2^t + \mu_1(\boldsymbol{\beta}^{\dagger t+1} - \bar{\boldsymbol{\beta}}^{\dagger t+1})$

18:     $\boldsymbol{v}_3^{t+1} = \boldsymbol{v}_3^t + \mu_2(\boldsymbol{\alpha}^{t+1} - \boldsymbol{M}\boldsymbol{\beta}^{\dagger t+1})$

19:     $t = t + 1$

20: **until** stop criterion is met

21: **return:** $\bar{\boldsymbol{\alpha}}^t, \bar{\boldsymbol{\beta}}^{\dagger t}$

---

## C EXPERIMENTS ON IMAGE CLASSIFICATION IN MULTIPLE INSTANCE LEARNING

### C.1 IMPLEMENTATION DETAILS OF BLACK-BOX MODEL

We defined the DeepSets permutation-invariant model (Zaheer et al., 2017) as a black-box model $f$ to be explained. According to (Zaheer et al., 2017), the model $f$ comprises two components: a representation function that transforms each instance, $\phi$, and a non-linear network that produces predictions from the extracted representation, $\rho$. We used ResNet-50 (He et al., 2016) pre-trained on ImageNet as the representation function $\phi$ and two-layer multi-layer perceptron (MLP) as the non-linear network $\rho$. Here, in $\rho$, we used the ReLU activation function for the first layer and the softmax function for the second layer. Also, the number of hidden units in the MLP was set to 1,024. The model first extracts the representation of each instance using $\phi$, then adds them up into a single representation, and finally, outputs a prediction by applying $\rho$ to the aggregated single representation. We trained the model using our MIL image classification dataset with Adam optimizer (Kingma & Ba, 2015) with a learning rate of 0.001, a batch size of 32, and a maximum epoch of 300. The test accuracy of the model was 0.945.

### C.2 ADDITIONAL QUANTITATIVE EVALUATION

Figure 6 shows the insertion scores of the estimated HiFAs over various numbers of perturbations for the LoFAs, $N_{\mathrm{L}}$, where we fixed the number of perturbations for the HiFAs, $N_{\mathrm{H}} = 5$. As with the deletion scores in Figure 2a, the proposed method consistently achieved the best insertion scores.

Figure 7 shows the NDCG, insertion, and deletion scores of the estimated HiFAs over various $N_{\mathrm{L}}$, where we fixed $N_{\mathrm{H}} = 20$. This result shows that the methods other than BU-LIME equally achieved the highest NDCG and insertion scores regardless of $N_{\mathrm{L}}$ because $N_{\mathrm{H}}$ was sufficiently large to estimate the HiFAs accurately.

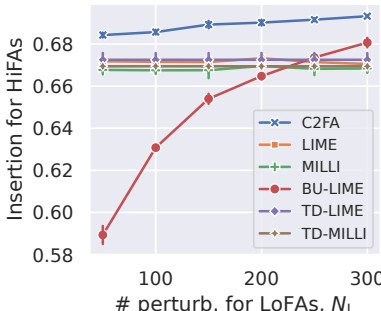

Figure 6: Insertion scores (higher is better) of the estimated HiFAs on the image classification task. The error bars represent the standard deviations of the scores over three runs with different random seeds.

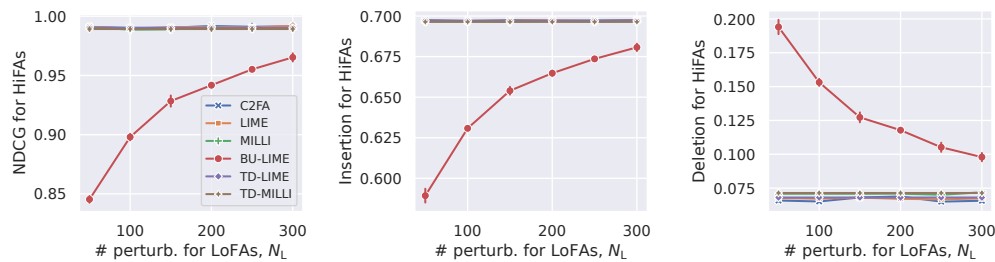

Figure 7: NDCG, insertion, and deletion scores of the estimated HiFAs on the image classification task when the number of perturbed instances $N_{\mathrm{H}}$ is 20.

### C.3 ADDITIONAL EXAMPLES OF ESTIMATED FEATURE ATTRIBUTIONS

Figure 8 shows additional examples of the estimated HiFAs and LoFAs by the proposed method and the best-comparing method, LIME, on the image classification tasks. In the setting where $N_{\mathrm{H}} = 20$ and $N_{\mathrm{L}} = 50$, the HiFAs tend to be estimated accurately, and the LoFAs tend to be estimated inaccurately because $N_{\mathrm{L}} = 50$ is small. Therefore, LIME tended to assign high LoFAs to incorrect regions. On the other hand, the proposed method was able to assign high LoFAs to correct regions by complementing the insufficiency of $N_{\mathrm{L}}$ with the accurate HiFAs.

## D  EXPERIMENTS ON TEXT CLASSIFICATION USING LANGUAGE MODELS

### D.1 ADDITIONAL QUANTITATIVE EVALUATION

Figure 9 shows the insertion scores of the estimated HiFAs and LoFAs over various values of $N_{\mathrm{L}}$ where we fixed $N_{\mathrm{H}} = 5$ and 50, respectively. With the insertion scores of the HiFAs, the proposed method became better than the MILLI-based methods as $N_{\mathrm{L}}$ increased, as with the deletion scores in Figure 4a. With the insertion scores of the LoFAs, the proposed method outperformed the other methods regardless of the values of $N_{\mathrm{L}}$.

Figure 10 shows the insertion and deletion scores of the estimated HiFAs over various values of $N_{\mathrm{L}}$ where we fixed $N_{\mathrm{H}} = 50$. In the setting where $N_{\mathrm{H}}$ is sufficiently large, the MILLI-based methods are superior to the LIME-based methods, including the proposed method, on the text classification task. This result suggests that the better approach for this task would be to formulate the estimators of the HiFAs (2) and LoFAs (3) with the MILLI-based sample weight kernel and optimize them simultaneously with the proposed consistency constraints. However, since no study has applied the idea of MILLI for estimating the LoFAs, we left the attempt for future work.

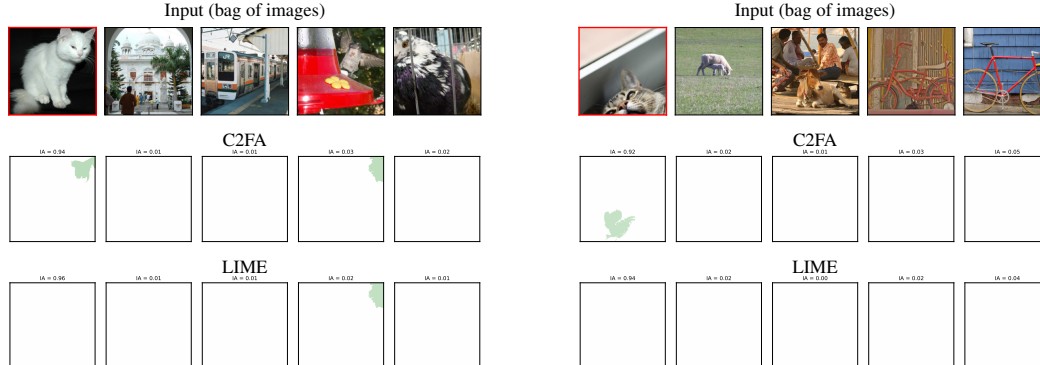

Figure 8: Two additional examples of the estimated HiFAs and LoFAs in the image classification task when $N_H = 20$ and $N_L = 50$. The input is shown on the first row, where the image with the red border is the positive instance. The HiFAs of super-pixels estimated by the proposed method (C2FA) and LIME are shown on the second and third rows, respectively, where the intensity of the green color indicates the magnitude of the LoFA. Also, the score at the top of each subplot indicates the value of the estimated HiFA for the instance.

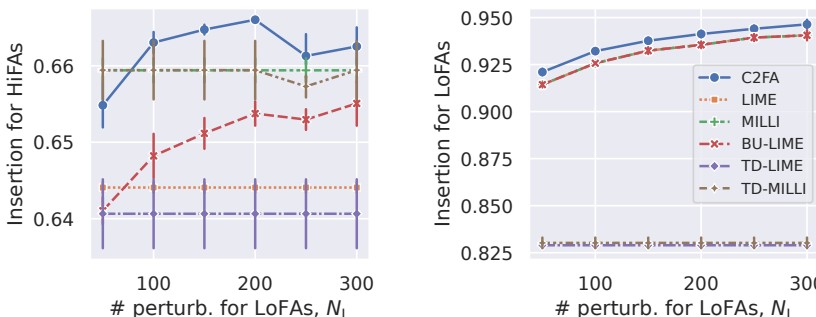

Figure 9: Insertion scores of (left) the estimated HiFAs and (right) the estimated LoFAs on the text classification task (higher is better).

## D.2 COMPUTATIONAL TIME

Figure 11 shows the average computational time of estimating the LoFAs using the proposed method on the text classification task when $N_H = 50$. From the figure, we can see that the computational time of the proposed method scales linearly with $N_L$ as with estimating the HiFAs and LoFAs separately. Here, the computational time of TD-LIME and TD-MILLI is constant against $N_L$ because they estimate the HiFAs only.

## D.3 ADDITIONAL EXAMPLES OF ESTIMATED FEATURE ATTRIBUTIONS

Figure 12 shows additional examples of the HiFAs and LoFAs estimated by the proposed method and the three comparing methods on the text classification task. The result shows that the sentence with the highest HiFA and the sentence associated with the word with the highest LoFA were consistent in the proposed method. However, the remaining methods did not show such consistency.

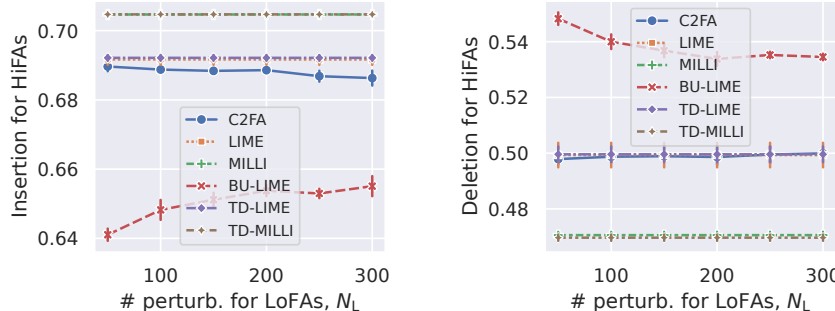

Figure 10: Insertion (higher is better) and deletion (lower is better) scores of the estimated HiFAs on the text classification task when $N_H = 50$.

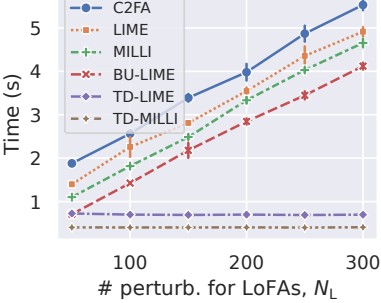

Figure 11: Average computational time of estimating LoFAs using the proposed method on the text classification task when $N_H = 50$.

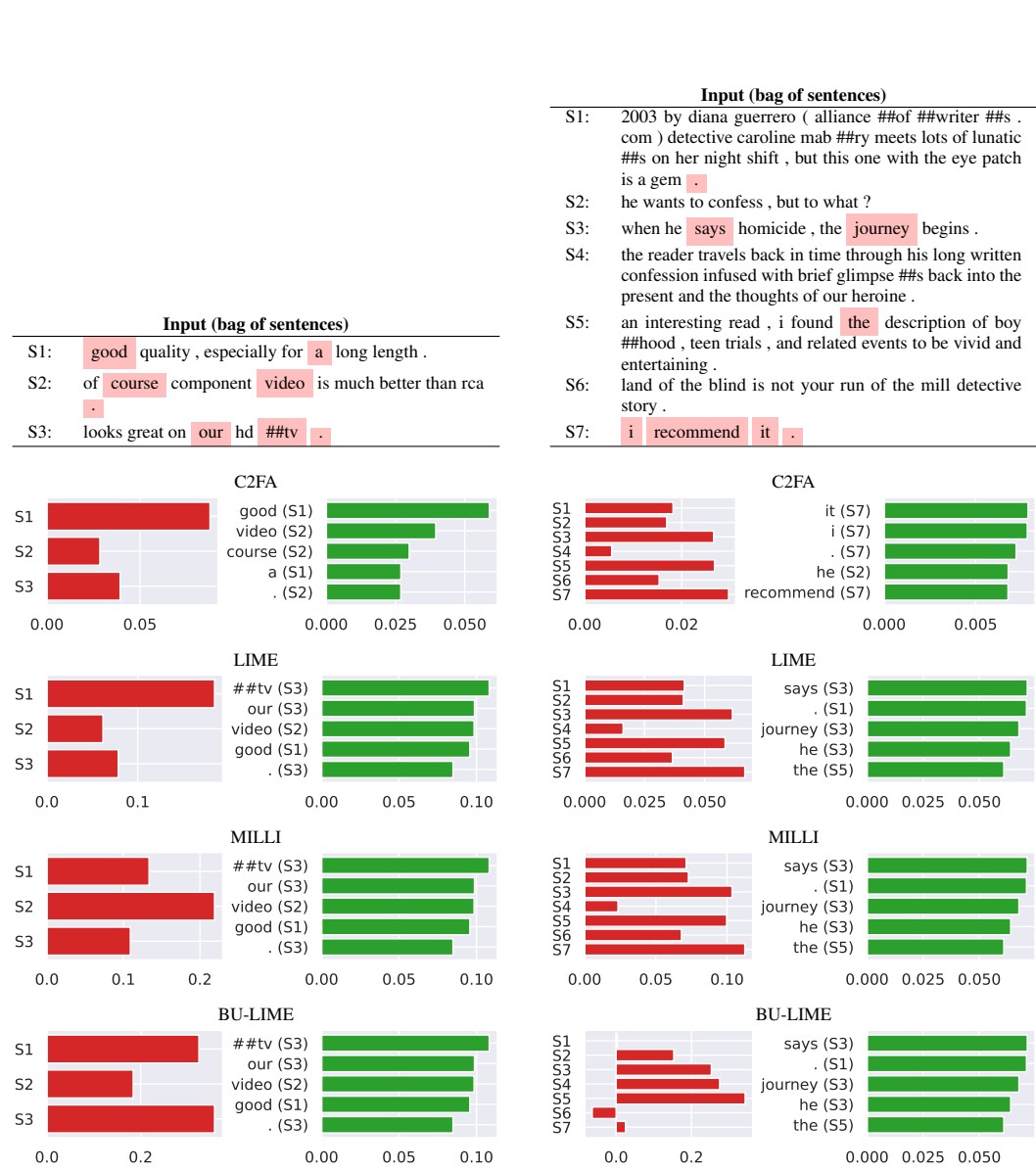

Figure 12: Examples of the estimated HiFAs and LoFAs for positive review texts when $N_{\mathrm{H}} = 50$ and $N_{\mathrm{L}} = 50$. The review text is shown at the top, and the HiFAs (left) and the top-5 highest LoFAs (right) estimated by each method are shown at the bottom. Here, the words on the pink background in the review text are those appearing in the chart of the LoFAs.