# OpenReview forum: "Explaining Black-box Model Predictions via Two-level Nested Feature Attributions with Consistency Property"
_ICLR.cc/2025/Conference — Submitted to ICLR 2025_

### Official Review · Reviewer_XLpk · 2024-10-15

**Soundness:** 2
**Presentation:** 3
**Contribution:** 2
**Rating:** 3
**Confidence:** 3

**Summary:**

This paper proposes a technique for extracting both high-level and low-level feature attributions from black-box machine learning models. The key idea is to simultaneously estimate high-level and low-level feature attributions while enforcing consistency between them through constraints, although they have not validated the assumption that high-level and low-level feature attributions show consistency for sure. The authors evaluate the method on image classification and text classification tasks, showing improvements over baselines in terms of attribution quality and query efficiency.

**Strengths:**

**Originality:**
- The idea of jointly estimating high-level and low-level feature attributions with consistency constraints is novel and well-motivated.

**Quality:**
- Experiments are comprehensive, evaluating on both CV and NLP tasks.
- Quantitative metrics assess different aspects like correctness, faithfulness, and consistency of the attributions.

**Clarity:**
- The paper is generally well-written with clear visualization.

**Significance:**
- The method enables more coherent explanations for nested input structures, which are common in many real-world ML applications.
- The approach is model-agnostic and can be applied to various types of nested inputs and black-box models.

**Weaknesses:**

- One foundamental assumption of this paper is that “ the input features have a nested structure that consists of high- and low-level features, and each high-level feature is decomposed into multiple low-level features. ”

 - Another important assumption is that there is consistency between low and high level features. However, what if conflicting attributions between high and low levels? For instance, when a high-level feature is deemed important, but none of its constituent low-level features appear significant. As the paper mentions that C2FA may perform worse when the consistency property is inherently not satisfied, I feel this is a more realistic consideration, that there is no consistency.

 - The paper lacks a formal theoretical analysis of the properties and guarantees of the proposed method. For instance, there is no discussion on convergence properties of the optimization algorithm or bounds on the estimation error, or discussion on the robustness, the robustness of C2FA to different types of perturbations or variations in the input structure.

 - It's unclear how the choice of hyperparameters (e.g., λ_H, λ_L, μ) affects the trade-off between consistency and individual attribution accuracy.

 - The experiments are limited to relatively small-scale problems (e.g., classifying a few images or sentences). It's not clear how well C2FA would scale to larger, more complex nested structures like full documents or large image collections.

 - The comparison with MILLI seems unfair, as MILLI is designed only for high-level attributions. A more appropriate baseline would be to combine MILLI with a low-level attribution method.

 - There's no comparison with more recent explanation methods like SHAP-based approaches or gradient-based methods adapted for nested structures.

 - The image classification experiments use a synthetic MIL dataset. It would be more convincing to see results on standard MIL benchmarks or real-world applications.

 - For text classification, only sentiment analysis is considered. Evaluating on more diverse NLP tasks would strengthen the claims of generalizability.

**Questions:**

- Just a brainfart thought, why 2 levels? How does the performance of C2FA change as the depth of the nested structure increases (e.g., to 3 or more levels)? Are there theoretical or practical limitations to extending beyond two levels? C3FA can make sense in an NLP context,  Word-level to sentence level to paragraph level.

- The paper mentions that C2FA may perform worse when the consistency property is inherently not satisfied. I feel this can be a more realistic setup, which makes C2FA less pragmatic, any justification here?

- The current formulation assumes a fixed nested structure. How might C2FA be adapted to handle variable-length or dynamic nested structures, such as those often encountered in NLP tasks?

- The experiments focus on classification tasks. How well do you expect C2FA to generalize to regression problems or other types of model outputs?

---

> ### Author Response · Authors · 2024-11-17
>
> Thank you for taking the time to review our paper and for your thoughtful comments. Below, we provide detailed responses to each of your questions.
>
> > Just a brainfart thought, why 2 levels? How does the performance of C2FA change as the depth of the nested structure increases (e.g., to 3 or more levels)? Are there theoretical or practical limitations to extending beyond two levels? C3FA can make sense in an NLP context, Word-level to sentence level to paragraph level.
>
> The decision to focus on two levels was driven by the practical applications we were targeting, such as multiple instance learning (MIL) and sentence-level text classification, where the input naturally follows a two-level nested structure (e.g., instances within bags or words within sentences). However, it is theoretically possible to extend our method to deeper nested structures (e.g., three levels: word → sentence → paragraph in NLP tasks) by introducing additional consistency constraints. In this case, we would incorporate extra consistency terms between the attributions at each level, ensuring alignment across all levels.
>
> While our current formulation is optimized for two levels, adding more levels would increase the complexity of the model, potentially leading to higher computational costs and making convergence more challenging. Nevertheless, by extending the consistency constraints to deeper levels, it is feasible to handle more complex nested structures. We plan to explore these extensions in future research to assess both the theoretical implications and practical scalability of our approach.
>
> > The paper mentions that C2FA may perform worse when the consistency property is inherently not satisfied. I feel this can be a more realistic setup, which makes C2FA less pragmatic, any justification here?
>
> You raise a valid point. Indeed, there are realistic scenarios where high-level feature attributions (HiFAs) and low-level feature attributions (LoFAs) may not always be consistent. For example, it is possible for a high-level feature to appear significant while its constituent low-level features do not, or vice versa. In such cases, enforcing the consistency constraint could potentially lead to suboptimal explanations. However, our method is designed to strike a balance between consistency and faithfulness. The consistency constraint is weighted using hyperparameters ($λ_H$, $λ_L$), allowing flexibility in scenarios where strict consistency may not be realistic. Additionally, monitoring the optimization loss during training can help detect when the consistency assumption is not being satisfied, allowing for adjustments. In future work, we aim to explore adaptive mechanisms that can dynamically adjust the importance of consistency based on the nature of the data.
>
> > The current formulation assumes a fixed nested structure. How might C2FA be adapted to handle variable-length or dynamic nested structures, such as those often encountered in NLP tasks?
>
> Our proposed method constructs explanations on a per-sample basis, which means it can naturally handle inputs with varying lengths and nested structures across different samples. Since C2FA estimates feature attributions for each sample independently, it is not limited by fixed-length inputs or rigid nested structures. Therefore, the applicability of C2FA to variable-length or dynamic nested structures largely depends on whether the black-box model used for predictions can accommodate such data flexibly. As long as the underlying model can handle inputs of varying lengths, C2FA can effectively generate consistent explanations by leveraging the flexibility of its surrogate model approach.
>
> > The experiments focus on classification tasks. How well do you expect C2FA to generalize to regression problems or other types of model outputs?
>
> Similar to LIME, our proposed method, C2FA, can be applied to both regression and classification tasks using the same formulation. The process of estimating high-level and low-level feature attributions remains consistent, regardless of whether the output is categorical (classification) or continuous (regression). The key difference lies in how the surrogate model interprets the predictions: in regression, the explanations would focus on the continuous output values rather than class probabilities. This flexibility allows C2FA to generalize across different types of model outputs without requiring significant modifications.

---

### Official Review · Reviewer_SvJo · 2024-10-31

**Soundness:** 1
**Presentation:** 2
**Contribution:** 1
**Rating:** 3
**Confidence:** 4

**Summary:**

- Existing studies have primarily focused on estimating either HiFAs or LoFAs, without addressing both levels simultaneously. Current naive approaches either generate a large number of queries or produce inconsistent HiFA and LoFA estimations.
- This paper introduces a model-agnostic local explanation method that effectively leverages the nested structure of input data to estimate both levels of feature attributions simultaneously.
- Experimental results in image and text classification demonstrate that the HiFAs and LoFAs estimated by the proposed method are accurate, faithful to the behavior of black-box models, and provide consistent explanations.

**Strengths:**

- Overall, the paper is well-written and well-organized, allowing readers to follow the narrative easily.
- The HiFAs and LoFAs estimated by the proposed method in this paper are consistent.

**Weaknesses:**

- Lines 297-305: The settings of TD-LIME and TD-MILLI are unclear. Additional explanation is needed to help readers understand their significance.
- This paper lacks a formal introduction to the evaluation metrics NDCG (line 327) and HIML (line 339).
- Figure 3 lacks an introduction to "IA," and the font size should be enlarged.
- The experiments in this paper are conducted solely on synthetic datasets and lack results on real datasets, such as medical imaging data.
- In Figure 5, BU-LIME demonstrates a higher level of consistency between LoFAs and HiFAs compared to C2FA.

**Questions:**

- Lines 89-91: The paper states that estimating HiFAs and LoFAs separately can lead to inconsistent explanations between them. How might this mismatch reduce users' trust in the model?
- Section 4.1: What would the explanation be if a bag contains more than two images featuring "cats"?

---

> ### Author Response · Authors · 2024-11-17
>
> Thank you for taking the time to review our paper. We appreciate your constructive feedback. Below, we provide responses to each of your questions.
>
> > Lines 89-91: The paper states that estimating HiFAs and LoFAs separately can lead to inconsistent explanations between them. How might this mismatch reduce users' trust in the model?
>
> When the high-level feature attributions (HiFAs) and low-level feature attributions (LoFAs) are estimated separately, there is a risk that the explanations they provide may not align with each other. For instance, if the HiFAs indicate that a certain image contributes significantly to the model's prediction, but the corresponding LoFAs do not highlight any meaningful features within that image, users may perceive the explanations as contradictory or unreliable. This inconsistency can lead to confusion, thereby reducing users' trust in the model, especially in critical applications such as healthcare or finance, where users rely on model explanations to make informed decisions. By ensuring consistency between HiFAs and LoFAs, our method aims to provide more coherent and trustworthy explanations​.
>
> > What would the explanation be if a bag contains more than two images featuring "cats"?
>
> In our experiments, it is common for a bag to contain multiple images featuring "cats." In such cases, our method estimates that the images containing cats will have high HiFAs. However, the values of these HiFAs will be distributed based on the probability output by the model for the label "cats." The LoFAs are then estimated in a way that is consistent with the HiFAs, ensuring that the explanations align across both levels. Since we cannot attach images here, we plan to include additional examples of such cases in the Appendix to illustrate how our method handles bags with multiple relevant images.

---

### Official Review · Reviewer_up2p · 2024-11-04

**Soundness:** 2
**Presentation:** 3
**Contribution:** 3
**Rating:** 5
**Confidence:** 3

**Summary:**

The authors propose a method to get high level and low level feature attributions for image classification in multiple instance learning and text classification with multiple sentences using language models. They use a local surrogate model in a LIME-like style, while optimizing for aggregating both high-level and low-level features for prediction. They also include a consistency constraint to enforce that all the attribution scores of low-level features that belong to a high-level feature sum up to that feature. They evaluate on Pascal VOC for image and Amazon reviews for text datasets, using faithfulness, correctness, and consistency metrics, and show that their method outperform baselines.

**Strengths:**

- The paper proposes to estimate HiFAs and LoFAs simultaneously, while previous works have only estimated them separately.
- The consistency property proposed by the authors is a reasonable property for HiFAs and LoFAs.
- The paper does experiments that ablate different important properties on both image and text datasets.
- The proposed method performed better than all baselines in all the metrics, and can get better attributions with a smaller number of perturbations.
- The method can be used in both text classification with multiple sentences (inputing as one input) and MIL where there are multiple input images.

**Weaknesses:**

- The high-level features are constrained to predefined image/sentence and cannot be dynamically chosen by the method.
- With the bottom-up baselines, we can actually convert any feature attribution method to the BU version of it. The paper only compare with different versions of LIME and MILLI. It would be more convincing that faithfulness and consistency cannot be achieved together with more baselines such as (BU-)SHAP, (BU-)RISE[1], IntGrad[2].
- Insertion and deletion are only proxy metrics for faithfulness. Although there is no absolute best metric for faithfulness, it would be good to clarify that they are proxies.
- The object segments in VOC dataset are ground truths for the objects, but not necessarily the model explanation. The model can be using the unintended features to make the prediction. For example, if the wolf always appear with snow in the dataset, the model can be wrongly using the snow for predicting the class fox, but it doesn't mean that attributing to the snow is a wrong attribution. It can be the correct attribution for a model that uses spurious correlation.

[1] Vitali Petsiuk and Abir Das and Kate Saenko. RISE: Randomized Input Sampling for Explanation of Black-box Models. BMVC 2018

[2] Mukund Sundararajan, Ankur Taly, Qiqi Yan. Axiomatic Attribution for Deep Networks. ICML 2017

**Questions:**

- line 237. Why does ADMM have the advantage of estimating $\mathcal{\alpha}$ and $\mathcal{\beta}$ independently even though there is the interdependence?
- line 301-303. Does it mean the LoFAs for their higher level feature attribution are just randomly selected based on the corresponding HiFA? What if you take the independently computed LoFA (without the constraint), and distribute each HiFA  proportional to the independently computed LoFA?
- How do bottom-up version of other methods like BU-SHAP, BU-RISE, BU-IntGrad do?

---

> ### Author Response · Authors · 2024-11-17
>
> Thank you for taking the time to review our paper. We appreciate your thoughtful comments. Below, we provide responses to each of your questions.
>
> > line 237. Why does ADMM have the advantage of estimating and independently even though there is the interdependence?
>
> The advantage of using ADMM in our method lies in its use of auxiliary variables to decouple the optimization of interdependent variables, such as high-level feature attributions (HiFAs) and low-level feature attributions (LoFAs). By introducing auxiliary variables and Lagrange multipliers, ADMM reformulates the joint optimization problem into separate subproblems for HiFAs and LoFAs. This allows us to solve for HiFAs and LoFAs independently in each step, while ensuring that they converge to a consistent solution through the consistency constraint. Essentially, even though HiFAs and LoFAs are interdependent due to the consistency requirement, ADMM leverages auxiliary variables to split the optimization into manageable parts, thereby enabling efficient and independent updates.
> The concrete algorithm is described in Appendix A in the supplementary material.
>
> > line 301-303. Does it mean the LoFAs for their higher level feature attribution are just randomly selected based on the corresponding HiFA? What if you take the independently computed LoFA (without the constraint), and distribute each HiFA proportional to the independently computed LoFA?
>
> Such an approach is indeed possible as a baseline. However, if the goal is to estimate HiFAs consistently using the derived LoFAs, a bottom-up approach like BU-LIME would likely be more effective. This is because BU-LIME inherently maintains consistency by aggregating LoFAs to obtain HiFAs, ensuring that the attributions align well between the two levels.
>
> > How do bottom-up version of other methods like BU-SHAP, BU-RISE, BU-IntGrad do?
>
> We did not compare our method with bottom-up versions of other methods like BU-SHAP, BU-RISE, or BU-IntGrad because our proposed approach is specifically designed for surrogate model-based methods like LIME. For a fair comparison, we focused on methods aligned with this framework. Notably, (Kernel) SHAP is also a type of surrogate model-based method similar to LIME, with the primary difference being in the use of a kernel function. Therefore, our proposed method, which is optimized for consistency between high-level and low-level feature attributions, would also likely work well with SHAP. However, for methods like RISE and Integrated Gradients, which do not inherently use surrogate models, additional modifications would be necessary to enforce the consistency property, making direct comparisons less straightforward.

---

> > ### Comment · Reviewer_up2p · 2024-11-25
> >
> > Thank you for the clarification. However, I'm not entirely convinced that BU-RISE, BU-IntGrad would not work directly since the bottom-up aggregation should automatically enforce the consistency. You can aggregate the IntGrad scores up for each image and then you will get the HiFAs from the LoFAs. This is the same as how you did BU-LIME.

---

### Meta-Review · Area_Chair_wk26 · 2024-12-15

**Metareview:**

This paper introduces a model-agnostic explanation method for simultaneously estimating high-level and low-level feature attributions (HiFAs and LoFAs) in image and text classification tasks. The method uses a LIME-like local surrogate model with a consistency constraint to ensure that low-level attributions sum to their corresponding high-level features. Evaluated on Pascal VOC (images) and Amazon reviews (text), the approach outperforms baselines in faithfulness, correctness, consistency, and query efficiency, demonstrating its effectiveness in providing accurate and coherent explanations.

The reviewers raised several concerns regarding the novelty and experiments presented in the paper, which I briefly summarize below:

1. One of the comments that the reviewer made is "How do bottom-up versions of other methods like BU-SHAP, BU-RISE, BU-IntGrad do?" The response was not convincing as mentioned by the author.

2.  The settings of TD-LIME and TD-MILLI also seem unclear.

3. Experiments are conducted on synthetic datasets and lack real-world datasets.  It would be more convincing to see results on standard MIL benchmarks or real-world applications.

4. The paper lacks a formal theoretical analysis of the properties and guarantees of the proposed method.

5. The comparison with MILLI seems unfair.

6. Evaluating on more diverse NLP tasks would strengthen the claims of generalizability.

**Additional Comments On Reviewer Discussion:**

Many comments are raised by the reviewers. The authors seem to have addressed this partly. Many responses were not convincing.

---

### Decision · Program_Chairs · 2025-01-22

Reject